# Young adults' experiences of using a young person's mental health peer support app: A qualitative interview study

Bethany Cliffe[1,2], Myles-Jay Linton[1,3]*, Zoë Haime[1,2], Lucy Biddle[1,2]

1 Population Health Sciences, Bristol Medical School, University of Bristol, Bristol, United Kingdom, 2 The National Institute for Health Research Applied Research Collaboration West (NIHR ARC West) at University Hospitals Bristol and Weston NHS Foundation Trust, Bristol, United Kingdom, 3 School of Education, University of Bristol, Bristol, United Kingdom

☙ These authors contributed equally to this work.
* mj.linton@bristol.ac.uk

## Abstract

Evidence suggests that digital peer support can be valuable for individuals struggling with their wellbeing, particularly those who do not feel able to or do not want to engage with other services. The current study explores the experience of young adults engaging with a digital peer support smartphone app. Interviews were conducted with 11 young adults aged 18–25. Reflexive thematic analysis was used and five themes were developed from the data: 1) Finding comfort in familiar and friendly digital spaces; 2) Developing coping and support skills through digital peer support; 3) The value of shared experiences; 4) Needing to 'pull your weight' but being scared of causing harm; 5) The limits of digital peer support. We found that participants valued the sense of community and feelings of relief and validation elicited from sharing relatable experiences with peers. They also believed they had developed skills in supporting themselves and others both within and external to the app. However, it was mainly perceived as a space for venting and may not succeed in delivering benefits beyond this, such as in reducing symptoms of poor mental health or helping people get to the root of issues. Moreover, participants reported a pressure to respond and anxieties around exacerbating someone's difficult feelings. Providing training and supervision to peers to help them feel confident and safe when supporting others may help to further the benefits of peer support, and a greater emphasis on boundaries within digital peer support may alleviate some anxieties and pressure.

## Author summary

There is limited research looking at how young people experience using digital peer support apps, so in this study we interviewed young adults to understand what they think about it and how it impacts them. We found that participants enjoyed aspects of peer support like it being an informal and friendly space, learning about mental health and coping skills, and feeling like their peers understand what they are going through. On the other

**Data Availability Statement:** Whilst we acknowledge and value the importance of open science practices, we do not believe it is appropriate in this case to share the interview

transcripts in a repository. Firstly, we did not receive ethical approval to do so, and consent was not given by participants for sharing the transcripts in this way. Secondly, participants in this study came from a relatively small community of users from a single app meaning there is a greater risk of identification. This is particularly true given the in depth conversations that included potentially identifiable details which are integral to the data and so are difficult to redact or anonymise without losing meaning. Finally, the conversations often pertained to sensitive discussions around the participants' experiences of struggling with their mental health, including references to specific traumatic events or having received certain diagnoses which are not appropriate for sharing. In addition to the risk of identifying participants, sharing raises the possibility of breaching participants' privacy and trust in us as researchers after disclosing personal and often distressing information with us. If you have any questions or are in doubt, please email research-governance@bristol.ac.uk.

**Funding:** LB and MJL received funding from Innovate UK (https://www.ukri.org/councils/innovate-uk/) and the National Institute of Health Care Research Applied Research Collaboration West (https://arc-w.nihr.ac.uk/). The funders did not play any role in the study design, data collection and analysis, decision to publish or preparation of the manuscript.

**Competing interests:** The authors have declared that no competing interests exist.

hand, challenges of peer support included being scared of saying the wrong thing, feeling pressure to help others, and not necessarily being able to solve problems compared to professional support. Based on these findings, when designing digital peer support things that could be helpful are providing training or supervision for peers to feel confident and safe support others, considering how to introduce boundaries to reduce the pressure felt to respond to peers, and consulting young people to make sure the space feels friendly.

## Introduction

Peer support consists of individuals with similar experiences or identities coming together to share thoughts, feelings, advice, and empathy [1]. Contemporary peer support originated in response to communities receiving poor treatment from mental health services [1], but has since evolved into varying formats with varying functions. This includes formal peer support facilitated by a professional within mental health services or organisations, or informal peer support via social networks, forums or self-facilitated groups [2]. Research has evaluated in-person and online peer support for self-harm [3], suicide [4], anxiety, depression, quality of life and physical health conditions [5], typically with positive findings regarding its efficacy and acceptability [6]. Peer support is still something people often turn to after being met with mistreatment, stigma or other poor treatment from professional services [3]. Those engaged in digital peer support often cite the anonymity and degree of separation as a key benefit, as they are not required to formally 'identify' themselves and so forgo the possible risks of disclosure [4]. Consequently, digital peer support has been highlighted as particularly valuable for those concerned about potential stigmatisation or embarrassment from accessing professional services [7]. Research with people aged 18 and above suggests that digital peer support provides spaces to hear things about the experience of mental health that are typically 'hidden' within society, which helps people to feel like they are not alone [8]. Digital peer support has also been highlighted as a space to develop connections and friendships with people who can relate to each other's experiences amongst participants aged 21+ [9].

Despite promising findings regarding positive impacts on wellbeing and social connectedness, there are also concerns around it exacerbating difficulties in some instances. For example users of a self-harm forum were surveyed (mean age 21), and whilst most perceived the forum positively, 11.5% of respondents reported an increase in self-harming behaviours resulting from learning more severe methods of self-harm, or feeling less need to stop after hearing others' experiences and self-harm seeming more normalised [10]. Similarly, it has been reported by young people that hearing other's experiences of self-harm through peer support can lead to the behaviour becoming 'entrenched' and more severe [11]. Also, some have reported feeling misunderstood by peers and receiving unhelpful advice from them [9].

There is a clear need to explore individual's experiences of engaging in digital peer support in more detail. So far, evaluations of digital peer support have typically focussed on platforms such as bulletin boards, Facebook, text messaging, web-based platforms, or smartphone applications (apps) specifically for psychosis [12] or substance use [13]. Given the ubiquity and accessibility of smartphone apps, particularly amongst young people [14], evaluating the role they can play in peer support more generally is important. Previous research has identified that the sense of community within a mental health peer support app can improve adolescents wellbeing [15]. However, participants were typically under 18 [15], and this is yet to be explored in detail amongst young adults.

The current study aims to investigate the experience of young adults engaging with Tellmi, a UK-based digital mental health peer support smartphone app for young people, using qualitative interviews and reflexive thematic analysis. Specific research questions were:

1. What are young people's experiences of using Tellmi?

2. What are young people's experiences of digital peer support?

3. What meaningful impact does Tellmi have on young people's wellbeing?

## Methods

### Tellmi

Tellmi (previously MeeToo) was developed in 2017 and is a free to use UK-based digital peer support app in which young people (aged 11–25) can anonymously discuss a range of topics covering mental health and wellbeing with other age-banded peers. Users can post and respond to others' posts within the primary feed and they can also access a directory of over 600 different wellbeing resources that includes signposting to various mental health charities and organisations, websites, helplines, etc. All posts and replies are pre-moderated for safety. Please see https://www.tellmi.help/ for more information. Tellmi engage in research activities involving their user-base to improve app design and extend its offering. For instance, recent work to develop provision for high-risk users and to improve signposting within the app [16]. It has been commissioned by the NHS, and a previous evaluation of Tellmi with younger children found that it helps young people to feel less alone and has a positive impact on their mental health [15].

### Study design

A researcher conducted semi-structured interviews between June and September 2023 with 11 Tellmi users aged 18–25. These took place over a telephone call (n = 1), a video call using Microsoft Teams (n = 9), or via the chat function within Microsoft Teams (n = 1). Whilst the data gathered from a chat-based interview as opposed to audio/video may be different, it is important to ensure the interviews are accessible to all who wish to take part. It has been identified that interviews using this format are acceptable and do not elicit data of a worse quality [17,18]. Interviews lasted between 22–51 minutes (M = 36.6, SD = 9.27). Tellmi played no role in data collection or analysis, but they were involved in facilitating recruitment. Additionally, Tellmi staff helped to design the topic guide, to ensure our data collection was informed by technical expertise.

### Recruitment

Recruitment occurred via convenience sampling within the Tellmi app where the study was advertised within the app feed between July–August 2023. App users could click on a link to an online information sheet outlining the study and researcher's contact details should anyone wish to discuss the study further. The online link also contained a brief demographics form including space to enter details to be contacted by the researchers to arrange participation. The demographic questions were to help ascertain the diversity and representativeness of the study sample. After discussing the study via email, participants were then directed to an online consent form. 11/12 (92%) of those with whom the study was discussed over email participated; one felt they would be too anxious to take part in an interview. A further 34 registered interest but did not respond to contact from the researcher. Inclusion criteria included being

aged 18–25, being a Tellmi user and having a good enough grasp of English to take part in the interview. There were no exclusion criteria, including anything relative to specific mental health conditions, recruitment was open to all app users.

The sample size for this study was constrained by practical considerations due to a restricted window of time for recruitment, and a relatively small population to recruit from. We do not know how many app users saw the advert, but we do know that roughly 6,000 people use the app monthly and therefore may have seen it. As the advert was programmed to appear as a post when scrolling within the primary feed, users may not have accessed that part of the app or scrolled far enough to see it, which may have posed barriers to recruitment. Nevertheless, our sample size conforms to key requirements in line with an information power approach whereby the richness of the data meant fewer participants were required to address the research question [19]. The participants were articulate and shared in detail their experiences with the app which led to high-quality data. Also, it has been suggested that fewer participants are required to develop an understanding of the target population when it is a specific group, in this case Tellmi users aged 18–25 [19].

As a thank you for their time, each participant received a £25 voucher.

## Ethical considerations

Ethical approval was granted from the University of Bristol Faculty of Health Sciences Research Ethics Committee [13726]. Participants completed an informed consent form in advance and also gave verbal consent during the interview. The participant who participated via the chat function of Teams provided consent in writing during the chat. Following the interview participants were provided with a debrief sheet listing sources of support should they wish to use them. Since participants were users of a mental health support app, a distress protocol (please see S2 File) was created should the interview process trigger distress [20], but fortunately, this was not needed during the course of the study.

## Procedure

The research team developed a topic guide to broadly explore participants' experiences of digital peer support (S1 File). Questions were open and exploratory so interviews could be guided by participants, and prompts were used to gauge perceptions of more specific elements of digital peer support (e.g., age appropriateness, benefits and challenges). Example questions included 'can you tell me about your experience of using Tellmi', 'can you tell me about your experience of the peer support within Tellmi', 'can you think of any benefits/challenges of peer support from your experience', 'can you tell me whether Tellmi had any impact on your wellbeing', and 'what did you think of the design of Tellmi'. All interviews were conducted by a researcher (BC) with experience of qualitative research with young people on sensitive topics. Interviews taking place over Microsoft Teams were recorded using the in-built recording functions, and the telephone interview was recorded using an encrypted laptop. After each interview, BC wrote field notes documenting her experiences of conducting the interview, including the dynamic with the participant, any noteworthy observations and emotional reactions to the interview.

## Data analysis

A third party company transcribed the interviews and the research team used reflexive thematic analysis to analyse the data [21,22]. During analysis, BC continually reflected on her own positionality, acknowledging researcher subjectivity as integral to the process [23]. Firstly, BC read all transcripts to become re-familiarised with the data as it was transcribed by a third

party. BC then coded all transcripts using the comments function on Microsoft Word, coding both semantically and latently, and inductively and deductively. The corresponding field notes were considered alongside each transcript to provide context and enhance understanding of the data [24]. To explore any further insights ZH independently coded two transcripts. BC compared the codes and found considerable overlap, with interpretations not previously considered being factored in to analysis. Codes were printed out and manually organised into themes in an iterative process where different organisations were considered until the final themes were settled on. This was discussed with LB and MJL throughout this process until the team were happy that the themes accurately represented the data. The transcripts were regularly checked to ensure the themes remained grounded in the data. During theme development, the research questions were held in mind, but this was flexible to allow unexpected findings to still be represented within the themes. For example, whilst we anticipated participants discussing the possibility of receiving unhelpful advice, we did not necessarily expect discussions around the fear of causing harm to others, but this felt like an important finding to report.

## Results

### Participants

11 young people participated, they were aged 18–24 (mean 20.3) and identified as women (n = 7), men (n = 2), non-binary (n = 1) and 1 preferred not to say. In terms of ethnicity, participants identified as White (n = 6), North African (n = 1), Black (n = 1), Pakistani (n = 1), and 2 preferred not to say. Most participants had been using Tellmi for at least a few months, whereas a couple had only been using it for a few weeks. We have no specific data on length of use, only what participants self-reported during interviews. Four participants reported using Tellmi at least once a day, three used it at least once a week, whilst the remaining four participants gave no specific answer regarding frequency of use.

Five themes were developed using reflexive thematic analysis, please see Table 1.

### Theme 1: Finding comfort in familiar and friendly digital spaces

The format of this digital peer support app is similar to X (formerly Twitter) due to its post and reply system. Participants typically found this format familiar and comforting:

> "I think it's just something we've grown up with nowadays. It's just like a mental health social media page really and I think it's a format which people are used to [. . .] it's like Twitter in a way, I think that helps us be able to use the app clearly as well and makes us feel safer I suppose, it's that familiarity" [P12].

Similarly, the design of the digital peer support app was described as "friendly and calming" [P8] due to its bright colours, fun graphics and the incorporation of artwork created by users.

**Table 1. Themes.**

| Themes |
| --- |
| 1) Finding comfort in familiar and friendly digital spaces |
| 2) Developing coping and support skills through digital peer support |
| 3) The value of shared experiences |
| 4) Needing to 'pull your weight' but being scared of causing harm |
| 5) The limits of digital peer support |

This was contrasted with more "clinical" in-person spaces "that can be rather off-putting for young people" [P5]. Peer support being perceived as more "informal" [P5] than a clinical space was highlighted as beneficial as "it just feels less intimidating and more friendly in a way" [P12].

This was also felt within the support received, with several participants discussing how hearing information or advice from a professional may be perceived as patronising, compared to feeling more personal and valuable coming from a peer:

> "Maybe young people are more likely to listen if the information is coming from other young people rather than, say, professionals who might give it in a condescending way perhaps. Obviously, that might not be the professional's intention but it's just the way it comes across to young people" [P5].

This may be linked to beliefs that professionals help as it is their job to, whereas peers are seen as more altruistic given that they offer support voluntarily. Several participants reported that this made them feel better compared to professional support which can make them feel more alone:

> "they're not trying to make you exactly feel this way, but you sometimes can feel really intimidated because they are professional and they are paid to help you, basically. However, you can definitely benefit from having one [a therapist], yes, 100%, but you still see it as someone who you know, who has to be there, which can make you feel even more kind of alone. However, with Tellmi [. . .] they don't have to respond the way they do so, whenever you get positive responses it makes you feel they wanted to do that for you [. . .] which is very positive" [P4].

> "I guess it's nice that because these people have like chosen to reply to you, you don't feel like you've forced someone [. . .] It's sort of given to you without asking them. It's very generous, which makes it feel a bit different as well in a good way" [P6].

This sense of digital peer support as being friendly, welcoming, informal and familiar left most participants feeling like "I never have to be on my own for as long as I use the app" [P12], indicating that they perceive it as something comforting and reliable.

## Theme 2: Developing coping and support skills through digital peer support

Several participants discussed the opportunities for skills development within digital peer support, describing it as a place to "learn more about human nature" [P3] and the varied experiences of mental health more broadly, helping them to be "more mindful about the experiences of others" [P2]. It also provided an alternative source to learn more about their own mental health outside of traditional discourses of mental health:

> "without having access to a formal diagnosis route it was useful when I was younger, rather than for example Google had just told me I'd got depression, but through the app you realise actually symptoms cross over and your life isn't kind of dictated by that" [P9].

Further, they reported that receiving practical advice from others who have been through similar things was particularly helpful for finding solutions to problems and developing coping skills:

"I think that when people relate to what you're going through sometimes they have more specific advice because they went through it [. . .] it's usually something that helped them as well, [. . .] that's more helpful because we can put it right into practice" [P7].

This was discussed in relation to general wellbeing and coping skills, as well as more specific circumstances such as tips for exam revision or for transitioning to university, where they felt that digital peer support "will definitely improve my experience and it will definitely help me mentally. It will also help me try and get settled into Uni with whatever issues I have, like I can say, 'hey, this happened to me today' you know, and I usually would get positive responses" [P4].

One hoped that "the more like advice you read and receive I guess that's going into your brain to sort of help in the future" [P6], suggesting that skills learnt through the app could help them manage longer term by having a bank of coping skills to draw from moving forwards.

As well as finding ways of helping themselves, a few participants also discussed learning how to listen to and support others who may be struggling:

"I think maybe it's helped me become a little better at listening and supporting [. . .] I've read other replies and thought, 'that's a really good way to comfort someone' and then maybe adopted it a bit myself" [P6].

This then enabled them to help people outside of the app, leaving participants feeling more informed about how to respond to others' issues:

"I think seeing the way that other people reply to each other's posts can be quite helpful in day-to-day life. If someone came to you with an issue you'd be more informed about the kind of things that maybe you should say to reassure the person or give them advice because of things that you've kind of seen on Tellmi already" [P9].

### Theme 3: The value of shared experiences

As noted, participants found this digital peer support app helpful for disclosing feelings in a familiar and friendly digital space. This seemed to be further facilitated by the validation and relief that came from relating to peers who they had shared experiences with, as this helped to normalise their experiences:

"I feel relieved. Relieved, I think, is the right term that I would use because it makes me feel like I'm not actually the only one feeling this way. Sometimes [. . .] I thought someone wouldn't feel that way because, for me, it's, I think, weird. Weird that I am feeling what I'm feeling, but once I post about it some people actually do feel the same way, so it's nice to know" [P2].

Relating to others formed the basis of social connection as it helped participants feel as though they were not alone in their struggle and that there was a community around them who understood:

"I recommend it to everyone to use it because it's basically like you feel a sense of community and you feel that you're not alone and that is the strongest message, one of the best messages someone with mental health issues can receive from anyone is that they are not alone and like, it's so positive" [P4].

Most participants believed that the anonymity of the app helped to create a space where app users felt comfortable sharing their experiences and subsequently seeking others who could relate, which had a positive impact on participants:

"Due to the anonymity again, we're all so much more open and honest and it's surprising how many of us have gone through similar experiences which we probably wouldn't have spoken about in real life. And so it feels like you're not alone. There's other people who have been in similar situations to you, they can talk from experience and vice versa. And it just makes you feel less lonely and more listened to, more optimistic about the future and stuff" [P12].

Moreover, one participant expressed how connections developed within the app were experienced as easier than in real life due to feeling less pressure around friendship maintenance:

"You can often see the same user popping up and as much as there's no sort of direct like kind of text thing like where you can like message individual users, being able to kind of consistently see the same people, it does feel like you've got friends, but not having to keep up the relationship" [P9].

## Theme 4: Needing to 'pull your weight' but being scared of causing harm

An unfortunate consequence of social connectedness was seen in a few participants also reporting a more burdensome duty of care to peers within the app. They discussed not knowing quite how to respond to a peer's post and feeling guilty around taking more out of the app than they were putting in:

"You're a bit more reluctant to reply to things if you can't sort of word it properly or you're a bit worried about how it will come across. And then you sort of feel like you're posting more than you're replying to, kind of like you're not pulling your weight" [P9].

This was particularly apparent for posts that may have contained content relating to trauma or less frequently experienced issues where participants felt unable to adequately respond:

"Situations like sexual abuse and some more complicated situations that honestly I don't know how to help with or I don't even connect with because luckily I've never experienced. So for me it's just like I don't know how to show support so I prefer not to say anything. But then I feel bad because I see that they don't have any replies and it's a bit tricky because you don't want that person to be without any support, but at the same time you don't know how to support them or what to say or how to help really" [P7].

This highlights an uncertainty about how to respond appropriately to peers in vulnerable situations. Consequently, professional input within the app still serves a necessary role:

"Sometimes when you are replying to other people's posts there is kind of worry that you've misjudged the situation or you've said something that maybe would've negatively impacted the like initial person [. . .] I know that goes through the moderating process to be approved anyway, but I do feel like there needs to be like [. . .] another person in the situation that's more professional to answer the worry that the first person posted. Because yeah, there's always sort of the worry that you're kind of not trained enough to reply" [P9].

The idea of not having received any training was a concern for other participants, and for some was evidenced in having seen unhelpful responses within the app:

"There's no like actual training from the people who are giving replies or anything. Sometimes I've seen replies that don't really like help that much or they don't really match what's being said. I kind of feel like that's maybe not helpful to that person at that moment" [P11].

This suggests that those engaging with digital peer support may feel pressure to respond but, due to a lack of training, may not always respond in an appropriate or helpful way—despite having the best intentions. This was reflected in a couple of participants reporting feeling ignored or misunderstood when they themselves have posted and received no replies:

"If you don't get any responses to the comment, it can be a bit of a slap in the face and that might be just because people don't know what to say. It's a bit like being left unread" [P5].

"I think there was one about two weeks ago that I didn't get a response and I was a bit like, 'Aw.' A bit disappointed [. . .] I think maybe I felt a little bit ignored. I wouldn't want to feel that really, but I think that this time I felt it" [P6].

Similarly, one participant reported the negative impact of having received an unhelpful reply:

"It kind of reinforced whatever the original post was about. So kind of for example saying that you felt like a burden because of X, Y and Z, if someone didn't really understand what you meant it sounded like they were almost agreeing with you. The cycle continued and then you were like, 'Yeah, actually I am a burden because of X, Y, Z'" [P9].

## Theme 5: The limits of digital peer support

Despite the favourable comparisons between digital peer support and more clinical, professional support made by participants in theme one, there was a sense amongst some participants that there are perceived limits to what digital peer support can offer. It was often described as a platform for "venting" [P5], "journalling" [P9], getting things "off your chest" [P8] and "just to let it all out" [P2]. In this way, participants used this digital peer support platform to expel their thoughts, feelings and issues, particularly when they felt unable to disclose their difficulties in real life:

"it's really just a safe place where I can talk about anything, even things which I've never really disclosed to anyone who actually knows me. So it's made me feel safe probably. It's probably added to my safety, being able to disclose those things without the fear of all the consequences of disclosing it in other settings" [P12].

The above participant reflected positively on this experience, feeling safe to disclose within the app compared to being fearful of possible consequences outside of the app. This was commonly reported with participants feeling unable to talk to friends and family, and worrying about being burdensome and "bothering my friends" [P6]. However, this could also be interpreted as the app presenting a holding space for difficulties that may not necessarily be resolved and remain seemingly unknown to loved ones or mental health professionals outside of the app. This was highlighted by the following participant who acknowledged the limits of what digital peer support can achieve:

"Like it can't really fix anything because it's just words on my phone really. So I would tell a friend about something that they could fix or help me do something with and although the app is reassurance and there are times that it does make me feel better, it is still words on my phone" [P6].

This was contrasted against other types of support that offer more person-centred and evidence-based advice for getting to the root of problems:

"[Digital peer support] is not as person-centred. I've done psychotherapy in the past and it's very person-centred. You grow a very specific relationship with the therapist. This is more informal. So I'd say if you want specific tailored advice, I'd maybe say explore other avenues of support just because Tellmi is not as person-centred. It's more generic advice and the people on it aren't professionals. They may be able to empathise because they've gone through some similar experiences but they're not professional in the same way that a CAMHS therapist would be or someone from your local mental health hub would be" [P5].

This suggests that digital peer support can provide reassurance and comfort but raises questions around the utility of 'letting feelings out' and these not being resolved.

## Discussion

### Summary of findings

The results of this study suggest that within the Tellmi app, young adults find comfort in the informal nature and familiarity of digital peer support, and value the personable and informed support from peers with shared experiences. They reported learning skills through peer support that enabled them to better help themselves and others, both inside and outside of the app. One of the key benefits of peer support in Tellmi was users feeling like they were not alone and feeling relieved knowing others had felt the same way. Conversely, they also felt a pressure and responsibility to support others within the app and did not always feel equipped or able to do so. Peers lacking any training meant that responses could sometimes be unhelpful and exacerbate negative thoughts and feelings. Finally, some participants felt that peer support was limited to a space for venting and getting feelings out, with limited utility in resolving issues beyond this.

### Comparison with wider research

Peer support originated in response to barriers accessing services, with the purpose of gaining insight and support from those with perspectives not necessarily shared by mental health professionals who have not 'lived' the experience [1]. This was corroborated in the current study with the value of bonding over relatable experiences highlighted throughout as meaningful and eliciting a sense of community. Participants reported feelings of relief, reassurance, validation, comfort and friendship following interactions within the app. This is a common finding with peer support [4,8,15] and may be particularly true within this young adult population who discussed sharing advice around exams, relationships and transitioning to college or university.

On the other hand, participants felt that support outside of these more practical life events was more limited in this format. It seemed as though the main function of digital peer support was a space for venting and releasing thoughts/feelings, rather than to necessarily resolve issues as it is not very person-centred or in-depth and peers were thought to primarily offer comfort rather than solutions. Consequently, the valuable role that professionals can play was also discussed by participants who felt that professionals could input in ways that peers cannot, for example providing advice that is evidence-based and better able to get to the root of problems.

An apparent contradiction arises here as participants highly valued having shared experiences with their peers and the positive impact that this had for them of feeling relieved and feeling less alone. There may be individual differences here in that, whilst difficulties accessing services remain a persistent and pervasive issue, peer support may provide comfort, normalisation of experiences, relief and reassurance [25], but for some this is not enough and so specialist support may be required. Future research could help to untangle this relationship by identifying who may benefit most from peer support and who may need further support.

It may be that venting helps to normalise and maintain an identity of distress for app users, as opposed to challenging their feelings in order to resolve them. Normalisation of feelings and experiences has been portrayed positively in the current study and the wider literature [26–28]. However, it may be that normalising distress prevents people from seeking support to resolve their difficulties. Interviews with medical students experiencing distress corroborated that normalising situations or symptoms is a barrier to help-seeking, particularly if they believe that peers are experiencing similar or worse [29]. Similarly, interviews with young people found that normalisation of depression was a significant factor in avoiding seeking professional support [30]. Following this, having feelings or experiences normalised within peer support may not actually be conducive to improving wellbeing and mental health, and may hinder seeking professional support.

Participants in the current study discussed how the shared experiences with peers particularly facilitated support, enabling them to share advice and learn coping skills from each other. This experience was found in a similar evaluation of peer support chats, in which participants described the feeling of becoming each other's therapists [9]. However, the drawbacks of this were highlighted in the current study whereby participants felt burdened by the responsibility of providing support. They were anxious about saying the wrong thing and possibly making someone's situation worse, an experience reflected by those who mentioned having received unhelpful (but well-meaning) responses. This has been reported elsewhere [9,31], and suggests that more opportunities for training around how to best support peers in these digital peer support spaces may be beneficial to help young people develop the skills and confidence in supporting others. An example of this is the #chatsafe campaign which helped young people feel more confident and safe discussing suicide online, as well as feeling more willing to support people struggling with suicidality [32]. Posts being well-moderated is also key to try and prevent unhelpful responses where possible.

It may be that this burden and perceived responsibility becomes more impactful over time, with an increase in negative affect having been observed in young adults providing peer support for extended periods of time [6]. Research has identified burnout and the stress of the role as challenges to youth peer support in face-to-face settings, with supervision and boundaries recommended to protect against this [33]. However, these principles may not directly translate to digital peer support where there may be no supervision in place and limited guidance around boundaries. Future research should explore the impact of the length of engaging with digital peer support on peers' wellbeing and perception of their role and responsibility. There may be some value in considering restrictions around how many posts peers can reply to as a way of introducing boundaries and reducing this pressure. Artificial intelligence could possibly be employed here to monitor users' frequency of replies and offer supportive strategies to help prevent peer burnout, and exploring this in future studies may prove insightful.

## Strengths and limitations

This study adds valuable knowledge around the experience of digital peer support amongst young adults, providing a balanced view of both the benefits and the limitations. This study

was part of a broader academic-industry collaborative project, and a key strength of this research is the impact from feeding the findings back to Tellmi; findings were presented to Tellmi with recommendations for changes/additions that could improve the experience of peer support within the platform. This included specific design changes not covered in this article, as well as more conceptual ones such as providing more training for peers and considering the possibility of supervision for those taking more of an active role in the community. This builds upon other work we have done highlighting key elements of digital peer support that can help to extend its offering to those experiencing suicidal thoughts [16]. It is also the aim that key principles from this paper could be applied to other digital peer support platforms. However, it is acknowledged here as a limitation that these findings relate to one specific platform, so may not be generalisable to all others. A further strength is that data were collected from app users with explicit and ongoing experience of peer support, rather than hypothetical or historic examples.

A limitation of this study was the convenience sampling method used. Particularly, participants were a small group who self-selected to take part and it may be that this group volunteered to participate due to having more favourable attitudes towards the app. Considerably more participants registered interest than took part, and unfortunately we do not know for sure why they did not continue to participation. Additionally, participants were mostly White and identified as women. Recruiting a larger sample size in future would help to ensure greater diversity so that the experiences of the broader population engaged with digital peer support could be represented. Similarly, purposive sampling would help to recruit participants who vary in their engagement with the app, i.e., those who post and reply regularly compared to those who are more 'passive' and view the app without necessarily engaging with peers, as well as longer and shorter-term app users. Whilst we perceive a strength of this study to be the varying formats of interviews offered, it may still be that some users may have reservations about taking part in interview-based research. Anonymous surveys within the app may be appropriate for these individuals and should be considered for future research.

Finally, no data were formally collected regarding participants' length or frequency of Tellmi use. Similarly, no data were collected regarding participants current wellbeing or any mental health conditions they may have had. This information may have influenced the findings, so having an understanding of this would provide helpful context for this study. Future research should consider collecting this data where appropriate.

## Implications

Lived experience is perceived as very valuable and should be championed within digital peer support services. Supporting young people to perform these roles successfully and confidently whilst also prioritising their own wellbeing is key. Digital peer support platforms providing training opportunities for peers to learn how best to respond to posts and providing supervision to support them in this process longer-term may be beneficial. One consideration for training may be around how to provide peer support with less emphasis on normalisation of experiences if this can lead to avoiding professional support amongst those who may require it, as evidence suggests. As mentioned above, the #chatsafe campaign is a great example of how training could be delivered within an app through short videos, animations and text which coach the young person on how to respond to people in distress whilst keeping safe themselves. This has been evidenced to increase confidence and self-efficacy when communicating with peers in distress [32]. However, this must be carefully considered as other research into digital peer support training has found that providing too much information in a short space of time may in fact decrease people's readiness to support peers [34]. Further, if some peers are

trained and some are not, this may lead to a power imbalance within the app in which those who are not trained may lack the confidence to engage compared to their peers. Developing training and supervision informed by peers would be important for ensuring it meets their needs and is implemented appropriately.

The responsibility of supporting others can be overwhelming and anxiety-provoking. In addition to providing training and supervision, digital peer support platforms should emphasise the importance of boundaries and devise ways of constructing these within apps. One recommendation is exploring limits on how many posts an app user can reply to within a certain time frame may help to ease some of the pressure. This is an important consideration when designing digital peer support to ensure that all users are kept safe.

Finally, digital peer support spaces presenting as informal and familiar was perceived as comforting and accessible for young people. This should be taken into consideration when designing digital peer support for young people to mimic platforms that are accessible to young people, such as social media. As technology is constantly evolving, the concept of familiarity may prove challenging. As online spaces develop further beyond forum-based platforms, ensuring they remain friendly and informal should be a priority to help young people still feel comfortable in them. For example, how to create familiar environments within virtual reality based peer support may be an important consideration as this evolves [2]. Again, involving young people in informing this will be essential here.

## Conclusion

This research sought to understand the experience of young adults engaged with digital peer support provided on UK-based young person's mental health app. Benefits included being comforted by shared experiences and learning coping/support skills, whilst challenges were fearing causing harm and feeling pressure to support others. Key implications of these findings are discussed.

## Supporting information

**S1 File. Topic guide.**
(DOCX)

**S2 File. Distress protocol.**
(DOCX)

## Acknowledgments

The authors would like to thank the young adults who participated in this study. LB and BC are partly funded by the National Institute for Health Research Applied Research Collaboration West (NIHR ARC West) at University Hospitals Bristol NHS Foundation Trust. This work was supported by the Elizabeth Blackwell Institute, University of Bristol and the Development and Alumni Relations Office, University of Bristol. This project was also partly funded by Innovate UK.

## Author Contributions

**Conceptualization:** Myles-Jay Linton, Lucy Biddle.

**Data curation:** Bethany Cliffe.

**Formal analysis:** Bethany Cliffe, Myles-Jay Linton, Zoë Haime, Lucy Biddle.

**Funding acquisition:** Myles-Jay Linton, Lucy Biddle.

**Investigation:** Bethany Cliffe.

**Methodology:** Bethany Cliffe, Myles-Jay Linton, Lucy Biddle.

**Supervision:** Myles-Jay Linton, Lucy Biddle.

**Writing – original draft:** Bethany Cliffe.

**Writing – review & editing:** Bethany Cliffe, Myles-Jay Linton, Zoë Haime, Lucy Biddle.

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
