## [Decision Letter · Decision Letter 0]

30 Apr 2024

PDIG-D-24-00115

Young adults’ experiences of using a young person’s mental health peer support app: a qualitative interview study

PLOS Digital Health

Dear Dr. Linton,

Thank you for submitting your manuscript to PLOS Digital Health. After careful consideration, we feel that it has merit but does not fully meet PLOS Digital Health's publication criteria as it currently stands. Therefore, we invite you to submit a revised version of the manuscript that addresses the points raised during the review process.

Please submit your revised manuscript within 60 days Jun 29 2024 11:59PM. If you will need more time than this to complete your revisions, please reply to this message or contact the journal office at digitalhealth@plos.org. Please include the following items when submitting your revised manuscript:

We look forward to receiving your revised manuscript.

Kind regards,

Haleh Ayatollahi

Section Editor

PLOS Digital Health

Journal Requirements:

Additional Editor Comments (if provided):

Reviewers' comments:

Reviewer's Responses to Questions

**Comments to the Author**

1. Does this manuscript meet PLOS Digital Health’s publication criteria? Is the manuscript technically sound, and do the data support the conclusions? The manuscript must describe methodologically and ethically rigorous research with conclusions that are appropriately drawn based on the data presented.

Reviewer #1: Yes

Reviewer #2: Partly

Reviewer #3: Yes

2. Has the statistical analysis been performed appropriately and rigorously?

Reviewer #1: N/A

Reviewer #2: N/A

Reviewer #3: N/A

3. Have the authors made all data underlying the findings in their manuscript fully available (please refer to the Data Availability Statement at the start of the manuscript PDF file)?

Reviewer #1: No

Reviewer #2: Yes

Reviewer #3: No

4. Is the manuscript presented in an intelligible fashion and written in standard English?

Reviewer #1: Yes

Reviewer #2: Yes

Reviewer #3: Yes

5. Review Comments to the Author

Reviewer #1: The authors conducted a qualitative study on users’ experiences of using a peer support mental health app. I appreciated the conciseness of the qualitative reporting, the language used, and the richness the themes generated. I consider the study quite merited.

Evaluating the study reach is a considerable matter for the recruitment. In the Recruitment section, could consider describing to how many users the recruitment ad was shown to; in what time period; and how many clicked on the ad? You also mention that the study had a relatively small population to recruit from. How big was the population of users who had the capability to seeing the ad? How many of them were shown the ad?

Regarding ethical approval, consider including the university name.

Please briefly characterize the topic guide used in the interview also in the main text. Also, do mention who conducted the interviews.

What role, if any, did the Tellmi-platform developers and founders have in planning, conducting and analyzing the data?

Understanding the design studied is highly important in the evaluation of human factors. In other words, the participants’ responses may be considerably influenced by the design. Therefore, please elaborate on features of the Tellmi-application in a dedicated section: What are its key features? Is it moderated? Is it free to use? Who is it for? How long has it been in operation? How many monthly active users does it have? Who has developed it? What previous studies have been conducted on it? What kind of instructions does it have for posting and responding? How is the design improved and how frequently? How does it compare to other services? What is it relationship with other services?

Add the research questions at the end of the introduction in a dedicated section.

The Discussion explains that the study engaged in user-centered design by reporting the findings to Tellmi. This presents an intriguing option for future studies and iterations. As a reader, I would appreciate reading more about this considering that the ethos of this study is to create services (apps) that serve the users the best. How did this study further this aim, besides providing insights? How to foster impactful translation of research from findings to design iterations? 

I do appreciate how the themes were structured and reported. However, would a brief table at the beginning of the results section help the reader keep them in mind? 

“sospecialist” appears to be a misspelling.

The study reports sampling as a limitation. This is also associated with the interview method used in the study – interviews appeal to certain subpopulation. What other methods could / should be used to explore the topic in the future? 

At the end of discussion, the word co-design is used. However, there is some literature to suggest that the users should rather have the role of an informants than co-designer. I would advice rewording, while retaining the thought that evaluating the intervention with the users is important.

Reviewer #2: This paper presents a timely and insightful exploration of young adults' experiences with a digital peer support smartphone app for mental health. The study's qualitative approach, utilizing reflexive thematic analysis, offers a good understanding of participants' perspectives and sheds light on the benefits and limitations of digital peer support in this demographic group. The identification of five key themes provides a structured framework for interpreting the findings, and the discussion offers valuable insights into the potential implications for enhancing the effectiveness of such interventions. However, further clarification on methodological details, validity, and reliability measures would strengthen the paper's rigor. 

The authors mentioned usage time - a couple of months or weeks - but they don't display data regarding daily frequency - every day, several times a day, all the time... This may have an impact on the results. Looking at the interview guide there is a lot of information that is left out of their analysis, for example, usage data. In addition, the psychosocial state of young people who use the application is also possibly different in terms of stress, anxiety, depression, psychological well-being, and their responses will certainly be different. 

The chat interview seems to be quite different in the type of information it collects, in the same way that a telephone call does not capture important non-verbal information. Nevertheless, some authors pointed that the use of online chatting for interviews is acceptable when video or online interviews aren't feasible; It's also suitable as an additional method if it complements the research and allows for diverse data sources (eg. https://www.ncbi.nlm.nih.gov/pmc/articles/PMC10401366/).

The use of reflexive thematic analysis is appropriate for qualitative data, but it would be helpful to understand how rigor was maintained throughout the analysis process. Were there any efforts to mitigate potential bias during the analysis, such as reflexivity or member checking? What steps were taken to ensure the validity and reliability of the findings? For instance, were any strategies employed to enhance the trustworthiness of the data, such as triangulation or prolonged engagement with participants?

The identification of five key themes is informative, but could the authors elaborate on how these themes were derived from the data? Did any unexpected findings emerge during the analysis?

In discussing the limits of digital peer support, could the authors provide more insight into why participants perceived the app primarily as a space for venting rather than for deeper exploration of mental health issues? I think a paragraph explaining/describing the Tellmi app is needed at the end of the introduction or in the methods, since this app has several resources available.

The suggestion to provide training and supervision to peers is interesting. Can the authors expand on what this training might entail and how it could be implemented effectively within the app? Are there any potential drawbacks or challenges associated with providing such training, such as resource constraints or concerns about over-professionalization? Consider potential power imbalances in the collaboration process between these profiles.

What are some potential avenues for future research based on the findings of this study? Are there any unanswered questions or areas requiring further exploration?

How might the insights gained from this study inform the design and development of future digital peer support interventions for young adults? Maybe the authors can state more clearly some potential recommendations for future design and implementation of digital peer support for youth mental health based on results from their work.

Overall, despite the small sample, this paper contributes valuable knowledge to the field of digital mental health interventions and invites further inquiry into optimizing peer support approaches for young adults.

Reviewer #3: Peer review April 26 2024 Young adults’ experiences of using a young person’s mental health peer support app:

a qualitative interview study

Thank you to the authors for this interesting contribution to the literature. The authors write an article about the use of digital peer support methods such as mobile applications in the context of youth with mental health conditions. They authors used an interview approach and a qualitative interview which is an appropriate and important methodology to probe deeper into the subjective experiences of the users of these platforms. It appears users find certain benefits of using peer support apps that fill a gap between access to care and finding someone who is more relatable to them (ie. a peer and not an authority figure). I believe this is a valuable finding and this paper highlights certains strengths and weaknesses of digital peer support from the point of view of the user which is important. 

Intro

With regards to the introduction, I think it is reasonably well written and provides many relevant references to ground the paper, however I do feel the authors could enhance the references in this section surrounding self harm and prevalence / risks / challenges in youth psychiatry, going beyond the peer support groups (eg. Is the prevalence higher among those in support groups compared to those not). I find some of the language in the intro rather repetitive. Several times the author makes the same point that apps can be helpful, but on the other hand it has downfalls. And that there is research but at the same time lacking in research. I think the author should consolidate these thoughts perhaps in a more complete paragraph that frames the arguments for, and then another paragraph with the downfalls and then make the case for why more research is needed, instead of doing this multiple times every few sentences. 

- The authors should consider spending more time in the intro also covering a background on reflexive thematic analysis 

Methods 

- Which researcher did these interviews? Was it only 1 researcher who did them all? 

- The different mediums such as chat vs audio vs video is a bit unconventional and should be addressed as a potential limitation

- The authors speak a lot in the intro about youth and peer support groups however the actual methods report recruiting subjects aged 18-25. These creates some disconnect between the intro and the actual trial. 

- It is not clear based on the methods whether or not the researchers were seeking users with any particular mental health condition or just any type of user on the platform, can the researchers please specify. 

- What is the general usership of the Tellmi app? Are there hundreds, thousands, or millions or users? The authors note that there was a small population to recruit from and I wonder how that is the case if the app has a large usership

- Ethics approval: you said university faculty … of which university? 

- You state verbal consent but one participant was via chat function? How did they provide consent?

- Is the distress protocol available in supplementary materials?

- The authors say 11 ppl participated but list 12 people in the gender break down

Results

- Line 211 grammatical confusion with the quote saying “they” and “my”

- The authors should be consistent about punctuation (the period) to either put the period after the [] or before the “ “ 

o Currently the authors go back and forth with this formatting and sometimes do not put a . at all 

- Line 318 the quote says “being left unread” but I think the participant actually probably said “on read” which is a popular term meaning someone read your message but did not reply. The transcription service may have mis heard this

- Line 329 authors say “despite favorable comparisons” … I think it is not really suitable to compare this with clinical care as they are totally different things. There is a place for clinical care and place for digital support, but they are so far removed from one another it’s hard to truly compare them. 

- Line 383 the authors use in consistent referencing style and should be fixed 

- Line 402 more inconsistency in the referencing style .. the authors could incorporate thoughts and text from those references and cite them in normal fashion 

-

6. PLOS authors have the option to publish the peer review history of their article (what does this mean?).

---

## [Decision Letter · Decision Letter 1]

19 Jun 2024

Young adults’ experiences of using a young person’s mental health peer support app: a qualitative interview study

PDIG-D-24-00115R1

Dear Dr Linton,

We are pleased to inform you that your manuscript 'Young adults’ experiences of using a young person’s mental health peer support app: a qualitative interview study' has been provisionally accepted for publication in PLOS Digital Health.

Best regards,

Haleh Ayatollahi

Section Editor

PLOS Digital Health

Reviewer Comments (if any, and for reference):

Reviewer's Responses to Questions

**Comments to the Author**

1. If the authors have adequately addressed your comments raised in a previous round of review and you feel that this manuscript is now acceptable for publication, you may indicate that here to bypass the “Comments to the Author” section, enter your conflict of interest statement in the “Confidential to Editor” section, and submit your "Accept" recommendation.

Reviewer #1: All comments have been addressed

Reviewer #2: All comments have been addressed

2. Does this manuscript meet PLOS Digital Health’s publication criteria? Is the manuscript technically sound, and do the data support the conclusions? The manuscript must describe methodologically and ethically rigorous research with conclusions that are appropriately drawn based on the data presented.

Reviewer #1: Yes

Reviewer #2: Yes

3. Has the statistical analysis been performed appropriately and rigorously?

Reviewer #1: Yes

Reviewer #2: N/A

4. Have the authors made all data underlying the findings in their manuscript fully available (please refer to the Data Availability Statement at the start of the manuscript PDF file)?

Reviewer #1: No

Reviewer #2: Yes

5. Is the manuscript presented in an intelligible fashion and written in standard English?

Reviewer #1: Yes

Reviewer #2: Yes

6. Review Comments to the Author

Reviewer #1: The authors have now revised the manuscript and provided point-by-point responses. Their responses are comprehensive and succeed in addressing the points I have raised in my initial review.

Reviewer #2: The authors have thoroughly addressed all points indicated in the peer review.

7. PLOS authors have the option to publish the peer review history of their article (what does this mean?). If published, this will include your full peer review and any attached files.

**Do you want your identity to be public for this peer review?** For information about this choice, including consent withdrawal, please see our Privacy Policy.

Reviewer #1: **Yes: **Lauri Lukka

Reviewer #2: None
